# Opiate Antagonists for Chronic Pain: A Review on the Benefits of Low-Dose Naltrexone in Arthritis versus Non-Arthritic Diseases

**DOI:** 10.3390/biomedicines11061620

**Published:** 2023-06-02

**Authors:** Praneet Dara, Zeba Farooqui, Fackson Mwale, Chungyoul Choe, Andre J. van Wijnen, Hee-Jeong Im

**Affiliations:** 1Osteopathic Medical School, Des Moines University (DMU), Des Moines, IA 50312, USA; 2Department of Biomedical Engineering, University of Illinois at Chicago (UIC), Chicago, IL 60607, USA; 3Lady Davis Institute for Medical Research, SMBD-Jewish General Hospital, 3755 Cote Ste-Catherine Road, Room F-602, Montreal, QC H3T 1E2, Canada; 4Medical Research Institute, School of Medicine, Sungkyunkwan University (SKKU), Suwon 16419, Republic of Korea; 5Department of Biochemistry, University of Vermont (UVM), Burlington, VT 05405, USA; 6Jesse Brown Veterans Affairs Medical Center at Chicago (JBVAMC), Chicago, IL 60612, USA

**Keywords:** pain, arthritis, osteoarthritis, cartilage, naltrexone, opiate, regenerative medicine

## Abstract

Chronic pain conditions create major financial and emotional burdens that can be devastating for individuals and society. One primary source of pain is arthritis, a common inflammatory disease of the joints that causes persistent pain in affected people. The main objective of pharmacological treatments for either rheumatoid arthritis (RA) or osteoarthritis (OA) is to reduce pain. Non-steroidal anti-inflammatory drugs, opioids, and opioid antagonists have each been considered in the management of chronic pain in arthritis patients. Naltrexone is an oral-activated opioid antagonist with biphasic dose-dependent pharmacodynamic effects. The molecule acts as a competitive inhibitor of opioid receptors at high doses. However, naltrexone at low doses has been shown to have hormetic effects and provides relief for chronic pain conditions such as fibromyalgia, multiple sclerosis (MS), and inflammatory bowel disorders. Current knowledge of naltrexone suggests that low-dose treatments may be effective in the treatment of pain perception in chronic inflammatory conditions observed in patients with either RA or OA. In this review, we evaluated the therapeutic benefits of low-dose naltrexone (LDN) on arthritis-related pain conditions.

## 1. Introduction and Global Impact of Pain in Medicine

According to the International Association for the Study of Pain (IASP), pain is defined as “an unpleasant sensory and emotional experience associated with, or resembling that is associated with, actual or potential tissue damage”. Pain can be either temporary or acute, but it can also last for a long time. Persistent chronic pain conditions last longer than three months. Chronic pain is very prevalent in the United States [1]. Based on a National Health survey, approximately 50 million adults report pain on most days [2]. Frequently, multifaceted treatments are required to manage chronic pain [3]. Due to the prevalence of painful conditions, the clinical use of opioids has risen in recent years [4,5,6]. Although opioids provide modest relief, prolonged use of opioids is associated with several side effects, including drug addiction [7,8]. Furthermore, many opioid trials do not sufficiently account for patient variability in the pathophysiology of pain [9]. More recently, alternative methods to combat pain have emerged, including the use of opioid antagonists at low doses [10]. While this review focuses on low-dose naltrexone, we have provided a summary of the current uses of various doses of naltrexone in Table 1. This review provides an overview of the experimental evidence and current knowledge of the physiological effects of low-dose naltrexone for the treatment of arthritis-related pain perception.

This narrative review of the literature examines a broad range of papers that discuss the potential therapeutic benefits of low-dose naltrexone in different biomedical contexts. We performed searches in PubMed using the search terms “Naltrexone”, “Low Dose Naltrexone”, and “chronic pain”, complemented with searches in Google Scholar to account for studies not covered in PubMed. Results were primarily filtered for date of publication (from 2012 to 2023). However, some earlier studies were also included to provide historical context for the current studies. References include the entire set of publications we retrieved, as well as other relevant sources pertinent to this literature review. The only language restriction was that we only selected papers written in English. Two authors conducted the initial search (P.D. and Z.F.), two authors examined whether these studies were appropriate for inclusion (A.J.v.W. and H.-J.I.), and all authors approved the final list of references. Our manuscript summarizes these peer-reviewed published studies, and we remain agnostic about the clinical efficacy of low-dose naltrexone.

## 2. Applications of Naltrexone in Opioid Use Disorder and Alcohol Use Disorder

Naltrexone (C_20_H_23_NO_4_; MW: 341.4; trade names: Vivitrol, Revia) is a semi-synthetic opioid developed in the 1960s as an alternative to naloxone for opioid addiction treatment. It was first approved by the Food and Drug Administration (FDA) in 1984 for the treatment of opioid use disorder. Subsequently, the use of naltrexone was approved for alcohol dependency in 1994. The drug is prescribed in the range of 50 to 100 mg per daily dose and has been traditionally used to treat alcoholism or opioid use disorders [11,12,13]. Naltrexone is structurally and functionally similar to the opioid antagonist naloxone, but it has a longer half-life and better bioavailability. Naltrexone is an opioid antagonist that blocks the analgesic and euphoric effects of opioids [14]. Originally, naltrexone was developed as an oral medication, but its effectiveness was diminished by poor adherence. To counter the non-compliance of patients, formulations were created in which naltrexone could be injected or implanted. These next-generation formulations of naltrexone exhibit reduced dependency and overdose mortality while increasing the likelihood that a patient will remain in treatment for opiate use disorder and alcohol use disorder [15]. Additionally, a large Swedish study concluded that naltrexone as a monotherapy is associated with a lower risk of hospitalization for alcohol-related causes and does not increase mortality during administration [16]. Naltrexone has clinical benefits and represents a viable pharmacotherapy for opioid use disorder and alcohol use disorder. Because its drug safety profile is well established, naltrexone could also be considered for other clinical conditions.

## 3. Pharmacotherapeutic Effects of Naltrexone Are Biphasic

Although naltrexone has clinical benefits for substance abuse disorders, there have been clinical concerns regarding potential side effects. One early concern was the idea that naltrexone would worsen feelings of anxiety and depression because it is an opioid antagonist. The expectation was that naltrexone would have negative effects on the mental health of patients with opiate use disorder. However, experimental studies showed that administration of naltrexone reduced anxiety and depression levels back to normal levels in patients with opiate use disorder that otherwise would have higher levels of anxiety and depression [17]. Therefore, naltrexone may have therapeutic benefits with minimal side effects.

The pharmacological effects of naltrexone may be subjected to complex physiological feedback mechanisms. Naltrexone exhibits a biphasic dose response (i.e., hormesis) where it works as an inhibitor at high doses and as an agonist at low doses [18] (Figure 1). Consequently, low-dose naltrexone (LDN) in amounts ranging from 1 to 5 mg appears to work differently than the standard dosage typically used for opiate use disorder and alcohol use disorder. LDN seems to work beyond opioid receptor antagonism and may modulate inflammatory mediators. Mechanistically, the mu-opioid receptor (OPRM1, Opioid Receptor Mu 1) is the activating component of a G protein-coupled receptor (GPCR) signaling pathway. OPRM1 changes its association with heterotrimeric G protein alpha subunits during chronic opioid administration. Specifically, the OPRM1 signaling pathway switches from an inhibitory G*i* to a stimulatory G*s* state. This switch results in prolonged action potential and leads to several clinical effects, including hyperalgesia, tolerance, and dependence [19]. It has been documented that LDN not only enhances but also prolongs opioid analgesia and leads to decreased tolerance and dependence [20,21,22]. As a result, LDN has been explored for use as an immune modulator in inflammatory, rheumatologic, and neurologic diseases. Frequently studied conditions include fibromyalgia, MS, Crohn’s disease, and arthritis. The primary scope of this review is to present recent scientific evidence that would support the use of LDN against arthritis-associated pain.

## 4. Pharmaco-Kinetics of Naltrexone

Naltrexone is a bitter-tasting white crystalline compound that is soluble in water to a concentration of about 100 mg/mL. The solubility of the compound permits ingestion as an oral drug or delivery via injection of small volumes to reach daily doses (from 5 to 100 mg) for the treatment of substance dependency. For low doses, naltrexone is used in the range of 1 to 5 mg per day. At lower dosing, naltrexone may act beyond opioid receptor antagonism and modulate neuro-inflammatory processes, including attenuation of glial cells [20,23]. When naltrexone is administered orally, it undergoes complete absorption rapidly, and approximately 96% of the drug is absorbed from the gastrointestinal tract [23,24]. Although the drug is well absorbed, oral bioavailability estimates range from 5 to 40%. Approximately 98% of the drug is metabolized through liver cytosolic dihydrodiol dehydrogenases into 6β-naltrexol and two other minor metabolites [25]. Plasma levels of naltrexone reach their maximum within one hour upon administration of regular (>5 mg) or low doses (<5 mg), even though divergent biological effects are observed at either dose due to hormesis.

The plasma half-life of naltrexone varies depending on the route of administration. After oral administration of naltrexone, it takes 4 h to reduce the plasma concentration to half. The major active metabolite, 6β-naltrexol, persists in the biological fluid for a longer time (~13 h) and contributes to the pharmacological response of the drug [23,24,25,26,27]. When the drug is injected intramuscularly, it takes 5 to 10 days for the concentrations of naltrexone and its metabolites to be reduced to half. Most of the naltrexone and its metabolites (50 to 80%) are excreted, primarily by the kidney. The renal clearance for naltrexone is relatively rapid (ranging from 30 to 127 mL/min) and suggests that the drug is primarily excreted by glomerular filtration. In comparison, renal clearance for 6β-naltrexol is much slower (ranging from 230 to 369 mL/min), suggesting that an additional renal tubular secretory mechanism is operative. Consistent with the efficient conversion of naltrexone to other metabolites, its urinary excretion accounts for less than 2% of an oral dose, while conjugated 6ß-naltrexol accounts for 43% of an oral dose. Hence, the excellent bioavailability of naltrexone is determined by efficient intestinal absorption, active metabolic conversion in the liver to its active form (6β-naltrexol), and passive removal by kidney filtration. The half-life of naltrexone can vary from four hours to 5 days depending on the route of administration, and renal clearance involves multiple steps.

## 5. Physiological Mechanisms of Action of Low-Dose Naltrexone

The fundamental reason for considering LDN in the treatment of chronic pain is that the biological effects of naltrexone do not conform to linear dosing or saturation curves. Rather, naltrexone has biphasic effects and follows hormetic principles, in which a full dose acts as an inhibitor but a low dose of a drug acts as a weak agonist [20,26,27]. The conventional physiological action of naltrexone is as a competitive inhibitor of the opioid receptor that blocks the euphoric and sedative effects of opioids as part of the endogenous opioid system. Naltrexone is thought to act as a competitive antagonist for the µ, κ, and δ opioid receptors (OPRM1, OPRK1, and OPRD1) in the central nervous system, with the highest affinity for the μ-opioid receptor (OPRM1). Naltrexone competitively binds to such receptors and may block the effects of endogenous opioids and antagonize the effects of exogenous opiates. However, low-dose naltrexone virtually behaves as an opioid receptor agonist and paradoxically acts as a pain-reducing agent.

Theoretically, a complete blockade of endogenous opioid systems would not be a desirable outcome with chronic pain patients because it would inactivate a powerful physiological reward system that modulates our behavior and psychological perspectives. However, LDN may reduce pain and act agonistically by triggering positive feedback mechanisms. LDN causes a transient opioid receptor blockade that then prompts the body to compensate for reduced receptor activity by upregulating both endogenous opioids and opioid receptors [28,29]. For example, LDN stimulates the production of the body’s endorphins, which represent natural opioids. Beyond opioids, LDN may also have immuno-modulatory effects by stimulating the production of enkephalin (derived from its precursor proenkaphilin, PENK), a messenger that commands the immune system to decrease the production of an inflammatory agent called Substance P (Tachykinin Precursor 1, TAC1) [30,31]. Hence, LDN (i) enhances endogenous analgesia by supporting opioid upregulation and (ii) suppresses inflammation by repression of critical immune factors and (iii) other mechanisms that are currently under investigation (see below). Therefore, LDN may potentially improve both health and quality of life for patients with chronic pain.

LDN may affect inflammation by modulating multiple immunological mechanisms. Apart from suppressing Substance P (TAC1), LDN has been shown to have anti-inflammatory effects on the central nervous system through glial cell regulation. LDN acts as a Toll-like receptor (TLR4) antagonist [20,32]. When TLR4 is activated in microglia, it stimulates the production and release of tumor necrosis factor α (TNF), interleukin-1β (IL1B), interferon-β (IFNB1), and nitric oxide [20]. This glial cell-related inhibition of TLR4 and the concomitant decrease in inflammatory markers by LDN provides another physiological principle that is of potential therapeutic benefit to patients with arthritis who experience dysregulated joint inflammation.

Consistent with this concept, the TLR4 inhibitor TAK-242 (Resatorvid) has been examined in in vitro and in vivo animal models of RA, where Resatorvid improved inflammatory symptoms (e.g., in joint tissues of arthritic rats) [24]. Although RA is a systemic inflammatory disease in which environmental risk factors play major roles, approximately 50% of the risk for RA development is attributable to genetic factors [25]. Despite this, inactivation of TLR4 does not necessarily diminish RA symptoms, as evidenced by studies showing that blockage of TLR4 does not yield a significant change in inflammatory cytokine levels in a placebo, double-blind, randomized study of RA patients nonresponsive to methotrexate [32]. LDN may to some degree be beneficial for arthritic patients by exerting anti-inflammatory effects mainly via the nervous system. Because pain and joint inflammation are physiologically distinct mechanisms that are biologically connected, a combination of drugs along with LDN may show benefits for arthritis treatments [26]. The promise of LDN for the treatment of arthritis has been recognized for more than three decades [33,34], but there has been a surprising paucity of studies dedicated to advancing the clinical potential of LDN in treating arthritis.

## 6. Therapeutic Potential of LDN in the Treatment of Arthritic Diseases

***Rheumatoid Arthritis and Seropositive Arthritis.*** LDN may have therapeutic potential in RA, which is an autoimmune condition characterized by chronic inflammation. Unlike OA, RA mainly affects the synovial lining of the joints rather than the articular cartilage [35]. LDN may also be considered for patients with seropositive arthritis (SA), which is observed in a subset of RA patients that carry specific antibodies (i.e., anti-cyclic citrullinated peptides; ACPAs) and are more likely to have nodules, vasculitis, and lung issues [36].

One interesting study assessed if there is a beneficial relationship between LDN use and significant changes in the dispensing of medicines for rheumatoid and seropositive arthritis [37]. This quasi-experimental study identified patients in Norway with rheumatoid or seropositive arthritis and involved comparisons with patients before and after the dispensing of medication. The patients were then stratified into three groups based on LDN exposure: one LDN prescription dispensed (LDN × 1), two or three LDN prescriptions dispensed (LDN × 2–3), and four or more LDN prescriptions dispensed (LDN × 4). Furthermore, the patients served as their own controls, with data collection before and after treatment. The results of the study showed that persistent use of LDN led to reduced dispensing of medications (e.g., disease-modifying antirheumatic drugs and NSAIDs) for rheumatoid and seropositive arthritis. While a limitation of this study is that it is not a randomized clinical trial, the promising results that were obtained will encourage further studies on the effects of LDN on different types of arthritis.

***Osteoarthritis.*** At present, there are no ideal disease-modifying drugs that treat both cartilage defects and pain in OA, which represents a disease involving the progressive breakdown of cartilage in articulating joints, most often affecting knees and hips. Initial therapy often includes anti-inflammatory drugs to reduce swelling and pain. LDN shows promise for the future in the treatment options for OA and has been considered in clinical trials (NIH clinical trials; NCT03008590). Ultra-low-dose naltrexone combined with traditional medications for the treatment of OA has shown better clinical outcomes than standard traditional medications [38]. Once the disease progresses to a severe chronic pain stage, more potent drug regimens using narcotics such as oxycodone may be considered.

To circumvent the major adverse effects of opiates, several novel treatment strategies have been developed. One very interesting pharmacotherapy utilizes Oxytrex, which represents a combination of oxycodone and ultra-low-dose naltrexone (ultra-LDN). In a three-week phase II clinical trial, Oxytrex was administered to OA patients with moderate to severe pain [38]. In this clinical trial, patients were placed in one of four groups for treatment: placebo, oxycodone four times a day (qid), Oxytrex (qid), or Oxytrex twice a day (bid). The treatment groups all received the same daily dose except for the bid group, which received half the amount of daily naltrexone compared to the Oxytrex qid group (0.002 and 0.004). Through the course of the 3 weeks, the daily dosage of oxycodone gradually increased in all treatment groups, whereas the amount of naltrexone remained constant. Participants recruited for this study had a wide age range (i.e., 18 to 70 years old) and had experienced moderate to severe pain caused by OA for at least the past three months. The results of the study showed that Oxytrex administered twice a day was significantly better than any other treatment in terms of the reduction in pain intensity, the quality of analgesia, and the duration of pain control each day. As far as adverse effects are concerned, all treatment groups have similar amounts of adverse effects, such as opioid-related side effects. These findings on Oxytrex indicate that the therapeutic hormesis-based effects of LDN as an opioid receptor antagonist may work favorably in the presence of opiates that act as opioid receptor agonists.

Subsequent studies on Oxytrex [39] revealed that patients taking Oxytrex reported less physical dependence than patients on oxycodone. Additionally, the study found that patients taking Oxytrex twice daily had decreased constipation symptoms, somnolence, and pruritus, all of which are known side effects of opioids such as oxycodone. This large, controlled study was able to show effective analgesia effects from Oxytrex with negligible withdrawal symptoms. The main focus of this study was the examination of the effects of Oxytrex in patients with low back pain. Because lower back pain emerges from the degeneration of cartilaginous intervertebral disks, the findings are potentially relevant in the context of OA, which is characterized by the degeneration of articular cartilage. Importantly, ultra-low-dose naltrexone combined with oxycodone demonstrated fewer side effects compared to just oxycodone. Therefore, this drug combination may provide the basis for new treatment regimens to mitigate back pain and may potentially be considered for OA. A summary of findings of LDN use in RA and OA is provided below in Table 2.

## 7. Therapeutic Potential of LDN in the Treatment of Non-Arthritic Diseases

***Fibromyalgia*.** The immunomodulatory effects of LDN have been hypothesized to provide benefits for patients with fibromyalgia, which represents a neuroinflammatory condition characterized by widespread musculoskeletal pain associated with sleep, memory, and mood issues. There is some evidence indicating that LDN can provide pain relief and improve quality of life in patients [40]. In one early study on LDN in fibromyalgia, LDN was found to mediate a modest reduction (~30%) in symptoms of pain, fatigue, and stress based on examination of the erythrocyte sedimentation rate as the primary marker for these conclusions [20,41]. Follow-up studies also reported significant decreases in inflammatory cytokines such as interleukin-6 (IL6) and TNF-α [20,30]. Studies on fibromyalgia patients have also considered physical pain, mental health, and physical functioning in the context of adverse effects due to LDN administration. At the end of the study, many participants (~50%) experienced a partial improvement in symptoms, while a smaller group of participants (~20%) had a major overall improvement in symptoms [26]. Despite these encouraging results, one important limitation of this study is that many of the participants in this clinical trial not only had symptoms of fibromyalgia, but also other autoimmune disorders. Nevertheless, it appears that LDN is clinically useful for the treatment of fibromyalgia due to reducing pain symptoms and inflammatory markers.

In a more recent study, the dose–response relationship when treating fibromyalgia with LDN was explored [42]. The study included 25 subjects aged 18 to 60 that were dosed with between 0.75 mg and 6 mg of naltrexone. The effective drug doses for either 50% or 95% of the desired pharmacologic effects were estimated at 3.88 mg (ED50) and 5.40 mg (ED95), respectively. As a secondary outcome, the study measured the effects of the ten most common fibromyalgia symptoms, including energy levels, musculoskeletal stiffness, waking unrefreshed, depression, concentration/memory, anxiety, tenderness to touch, imbalance, and sensitivity to sensory inputs. These symptoms were evaluated via a questionnaire and each symptom was scored on a scale from 0 to 10 in terms of severity during the last seven days of the study. The symptoms with the greatest improvement in the questionnaire were found to be ‘tenderness’ and ‘waking refreshed’, with a mean change of −2.3 and eight patients reporting a 30% improvement in symptoms. All responders had at least a minimum improvement of 30% for at least one of the 10 FM symptoms, while most of the participants showed a minimum level of improvement (30%) for several symptoms. Although LDN does not mitigate most symptoms of fibromyalgia, the beneficial effects experienced by the patients indicate that LDN is sufficiently effective to improve quality of life. Overall, it appears that LDN has potential in the treatment of fibromyalgia due to both reducing pain symptoms and altering pain quality, while the overall safety and tolerability of LDN will not decrease general quality of life.

***Multiple Sclerosis.*** LDN is a possible treatment option for MS, a chronic and autoimmune disease of the central nervous system that manifests itself as either progressive or relapse-remitting MS [43]. Although the exact etiology of MS remains to be further established, the disease is associated with astrocyte inflammation leading to the recruitment of T cells to the CNS, demyelination, axonal damage, and neurodegeneration [44,45].

A limited number of studies have examined the therapeutic effects of LDN in patients with MS. In one retrospective chart review [46] after LDN therapy, 75% of patients noticed increased or stabilized quality of life. Likewise, in a case report, an LDN dose of 4.5 mg was provided and the patient had a decrease in the frequency and duration of migraine headaches as a result [47]. In another study, serum [Met5]—enkephalin (OGF/PENK) levels were examined as a biomarker for MS. The study found that patients diagnosed with MS relative to non-MS neurologic patients had reduced serum levels of OGF, which is an inhibitory peptide that suppresses the proliferation of T and B cells and their respective cytokines [48,49]. Furthermore, LDN therapy either administered alone or with Copaxone restored the patient’s enkephalin levels. Thus, LDN may be considered for the treatment of MS because it may correct imbalances in the immune system and improve the quality of life of MS patients.

***Crohn’s Disease.*** LDN has also been used as a novel treatment for Crohn’s disease, which is a multifaceted inflammatory condition affecting the digestive tract. Deregulated opioid signaling is often associated with Crohn’s disease, and this deregulation influences the secretion and motility of the gut. Recent studies have indicated the benefits of LDN for patients with Crohn’s disease (Table 3).

## 8. The Relationship between LDN and Depression in Chronic Conditions

The majority of the chronic diseases discussed here have a significant impact on both the physical and mental health of patients. A previous review that examined 38 studies on OA noted that both anxiety and depression are highly prevalent in patients with OA. Patients diagnosed with these conditions experienced more pain and less optimal outcomes [53]. The relationship between pain and depression has been well documented and is significant in chronic conditions because pain can amplify feelings of depression and vice versa, leading to conditions that are difficult to treat [54]. Medication has profound effects on this reciprocal relationship between pain and depression. It has been demonstrated that participants with chronic pain and depression have significantly fewer benefits from antidepressants compared to individuals without chronic pain. Similar to OA, depression is a risk factor for RA, and if patients develop the disease, the course of the illness will then be more detrimental [55].

The difficulty in treating chronic pain conditions along with depression has led to novel approaches, including the use of LDN. In a small randomized controlled trial involving patients with depression, LDN was added to a treatment regimen involving dopaminergic antidepressants. The results showed a significant decrease in the Hamilton Depression Rating Scale [56]. Although larger studies are needed to establish a clear benefit in treating depression, the results are promising, and the mechanism of action may involve dopaminergic receptors [23]. Further, a recent case report was released in which LDN was added to the treatment regimen of a patient with fibromyalgia and depression, resulting in an increase in quality of life, relatedness, and motivation [57]. Similarly, to fibromyalgia, patients with MS who were prescribed LDN or who had LDN added to their treatment protocol exhibited a decrease in levels of anxiety and depression [58]. While the relationship between chronic pain and depression has been well documented, new strategies to treat chronic inflammatory states may benefit from LDN. A summary of the benefits of LDN in non-arthritic causes is provided below in Table 4.

## 9. Strengths and Limitations

This is a mini literature review on the benefits of LDN in arthritis-related pain conditions and other pain conditions. LDN is an emerging treatment option, and this paper reviews some of the most recent evidence encouraging the use of LDN. At present, there is an insufficient number of double-blinded clinical studies that examine the utility of LDN in various chronic pain conditions to develop a definitive treatment plan for LDN. In addition, there is a very limited number of studies on the benefits of LDN in arthritis and firm conclusions are not yet possible. Our review merely provides a summary of relevant peer-reviewed published studies, and we remain agnostic about the clinical efficacy of low-dose naltrexone. Despite this, we are optimistic that additional fundamental and translational studies, as well as clinical trials, will be informative and may establish both the mechanistic basis and efficacy of LND in the future.

## 10. Conclusions

The clinical use of LDN provides relief in chronic conditions such as arthritis-related diseases and other inflammatory conditions, including fibromyalgia, MS, and Crohn’s disease. The present state of research on these medical conditions focuses on improving quality of life by reducing common symptoms. Future studies should expand on the symptoms and focus on disease-related biomarkers specific for each inflammatory disorder to supply clinicians with a more accurate way of providing diagnosis and treatment. For some inflammation-related diseases (e.g., MS), there are very few published studies at present. However, the currently available studies demonstrate both an improvement in symptoms and a decrease in serum markers. The future of treatment for MS using LDN is promising, but more studies are needed to demonstrate efficacy, safety, and replicability. For Crohn’s disease, LDN has proven to be safe while also improving the quality of life of patients. As with MS, larger clinical studies would help establish a clear protocol for the treatment of Crohn’s disease. LDN has been one of the most exciting novel treatment options for different types of arthritis. The limited number of published studies indicate that LDN decreases pain in patients and has few side effects. To further encourage LDN use in arthritis, more large-scale double-blinded studies are needed to examine the role of LDN. LDN shows promising use for future patients, but investments in clinical trials are required to investigate whether this treatment has significant benefits for patients with arthritis.

## Figures and Tables

**Figure 1 biomedicines-11-01620-f001:**
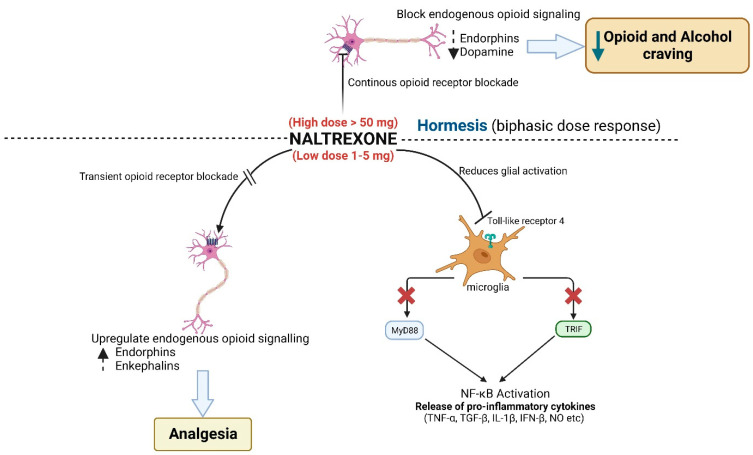
Schematic of the dose-dependent biphasic mechanism of action of naltrexone.

**Table 1 biomedicines-11-01620-t001:** Recent studies using naltrexone (including LDN and ultra-LDN).

Naltrexone Dosage	Clinical Uses
50–300 mg (regular dose)	Opiate, Alcohol addiction
1–5 mg (LDN)	Fibromyalgia, arthritis, Crohn’s disease, multiple sclerosis
0.001–1 mg very low-dose naltrexone (VLDN)	Reduced side effects and withdrawal symptoms of opiates
<0.001 mg (ULDN)	Similar to VLDN, OA, low back pain, postoperative pain control

**Table 2 biomedicines-11-01620-t002:** Summary of low-dose naltrexone’s therapeutic benefits in arthritis.

Disease	Characteristics of the Study	LDN Benefits
RA and seropositive arthritis	Quasi-experimental study with controlled before and after comparisons	Reduced dispensing of medications (antirheumatic agents and NSAIDS)
OA	A clinical study with Oxytrex (opiate+LDN) with a naltrexone dose of 0.004 mg	Reduction in pain and less dependence compared to opioids

**Table 3 biomedicines-11-01620-t003:** Recent studies involving LDN for Crohn’s disease.

Number of Participants	Length of Study, and Dosage	Outcomes of Study	Reference
17 participants	12 weeks, 4.5 mg naltrexone/day	Crohn’s disease activity index (CDAI) scores decreased significantly; 67% achieved remission (*p* < 0.001).	Smith [50]
47 participants	12 weeks, 4.5 mg naltrexone/day	LDN led to clinical improvement in 74.5% and remission in 25.5% of patients. Naltrexone improved epithelial barrier function by improving wound healing.	Lie [51]
Two studies were evaluated:34 adult patients,12 pediatric patients	12 weeks, 4.5 mg naltrexone/day 0.1 mg/kg up to 4.5 mg	LDN was safe and had minimal adverse effects.	Segal [52]

**Table 4 biomedicines-11-01620-t004:** Summary of low-dose naltrexone’s therapeutic benefits in non-arthritic diseases.

Disease	Characteristics of the Study	LDN Benefits
Fibromyalgia	LDN doses of 3.88 mg or 5.40 mg were used.	Reduces pain intensity and quality
Multiple sclerosis	LDN dose of 4.5 mg was used in one study. In another study, inhibitory peptides were indirectly measured as a way of assessing efficacy.	Reduces suppression of lymphocytes and improves quality of life
Crohn’s disease	LDN dosage of 4.5 mg naltrexone/day.	Wound healing, minimal adverse effects, improved symptoms
Depression and anxiety	Small-scale studies based on subjective assessments for patients with treatment regimens supplemented with 1mg LDN (bid).	Decreased levels of depression and anxiety

## Data Availability

Not applicable.

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
