# Peer review of "Opiate Antagonists for Chronic Pain: A Review on the Benefits of Low-Dose Naltrexone in Arthritis versus Non-Arthritic Diseases"

_biomedicines, 2023, doi:10.3390/biomedicines11061620_

Round 1

Reviewer 1 Report

This study entitled “Opiate Antagonists for Chronic Pain in Arthritis: Benefits of Low Dose Naltrexone” seems to have been generally well executed and written. Furthermore, I believe that this paper will be of great interest to the readers. Finally, I have only one minor remark and one major remark that require authors attention.

Title

Please include the type of your study in the title.

Methods

Please include this section in your work. State what the type of review you have performed (e.g., narrative review). Furthermore, briefly state what databases were screened for available literature, in which time period, was there any language restrictions during the search, who performed the search (initials of the authors), and which author (initials) has resolved any possible disagreements and approved the final list of included studies in your research.

Author Response

Reviewer #1 Comments:

“This study entitled “Opiate Antagonists for Chronic Pain in Arthritis: Benefits of Low Dose Naltrexone” seems to have been generally well executed and written. Furthermore, I believe that this paper will be of great interest to the readers. Finally, I have only one minor remark and one major remark that require authors attention.”

Reply: We thank the reviewer for these supportive comments.

Point 1:Title: Please include the type of your study in the title.”

We have modified the title to address this point. The amended title is: “Opiate Antagonists for Chronic Pain: A Review on The Benefits of Low Dose Naltrexone in Arthritis versus Non-Arthritic Diseases”.  

Point 2:Methods: Please include this section in your work. State what the type of review you have performed (e.g., narrative review). Furthermore, briefly state what databases were screened for available literature, in which time period, was there any language restrictions during the search, who performed the search (initials of the authors), and which author (initials) has resolved any possible disagreements and approved the final list of included studies in your research.”

Reply: In our experience, narrative reviews typically do not have a formal methods section. However, we have addressed this point by inclusion of an introductory paragraph that clarifies our methodology (i.e., we have not created a separate heading).

We have modified the paper to reflect how we performed our literature survey and have inserted the following paragraph:

This narrative review of the literature examines a broad range of papers that discuss the potential therapeutic benefits of Low Dose Naltrexone in different biomedical contexts. We performed searches in PubMed using the search terms “Naltrexone,” “Low Dose Naltrexone,” and “chronic pain”, complemented with searches in Google Scholar to account for studies not covered in PubMed. Results were primarily filtered for date of publication (from 2012-2023). However, some earlier studies were also included to provide historical context for the current studies. References include the entire set of publications we retrieved, as well as other relevant sources pertinent to this literature review. The only language restriction was we only selected papers written in English. Two authors conducted the initial search (PD and ZF), two authors examined whether these studies were appropriate for inclusion (AJvW and HJI) and all authors approved the final list of references. Our manuscript summarizes these peer-reviewed published studies and we remain agnostic about the clinical efficacy of low-dose naltrexone

Reviewer 2 Report

Following the analysis of the manuscript titled "Opiate Antagonists for Chronic Pain in Arthritis: Benefits of Low Dose Naltrexone", I appreciate the article's topic is interesting but it requires major improvements before considering publication and I recommend that it should be revised taking into account the following observations:

-          The title should be changed as it does not match the discussion about non-arthritic conditions.

-          Abstract: Please clarify the type of manuscript and what was the purpose of this article.

-          Introduction is very poorly documented. Provide several recent references for each of the following statements "Frequently, multifaceted treatments are required to manage chronic pain. Due to the prevalence of painful conditions, the clinical use of opioids has risen in recent years. Although opioids provide modest relief, prolonged use of opioids is associated with several side effects, including drug addiction. Furthermore, many opioid trials do not sufficiently account for patient variability in the pathophysiology of pain. More recently, alternative methods to combat pain have emerged, including the use of opioid antagonists at low doses."

-          Insert two tables summarizing the main characteristics of the clinical studies describing the therapeutic potential of LDN in the treatment of arthritic and non-arthritic diseases.

-          Before Conclusions, please clarify the strengths and the limitations of this narrative review.

-          The manuscript should be based on presenting especially the latest evidence from the chosen topic. Therefore, update the references because too many articles in the list have been published for more than 10 years (1, 2, 11, 20, 26, 28, 35, 36, 44, 47), even 15-20 years  (3-5, 13-18, 21, 30, 31).

-          What is reference 41??

Author Response

General commentary: “Following the analysis of the manuscript titled "Opiate Antagonists for Chronic Pain in Arthritis: Benefits of Low Dose Naltrexone"I appreciate the article's topic is interesting but it requires major improvements before considering publication and I recommend that it should be revised taking into account the following observations.”

Reply: We appreciate the constructive recommendations of the reviewer to modify our manuscript.

Point 1: “The title should be changed as it does not match the discussion about non-arthritic conditions.”

Reply: Similar to Reviewer#1, we have reworded our title to be more exact: “Opiate Antagonists for Chronic Pain: A Review on The Benefits of Low Dose Naltrexone in Arthritis versus Non-Arthritic Diseases”.  

Point 2: “Abstract: Please clarify the type of manuscript and what was the purpose of this article.”

Reply: This is a literature review that will be evaluating the therapeutic benefits of Low-Dose Naltrexone (LDN) in arthritis and non-arthritis related pain conditions.

Point 3: “Introduction is very poorly documented. Provide several recent references for each of the following statements "Frequently, multifaceted treatments are required to manage chronic pain. Due to the prevalence of painful conditions, the clinical use of opioids has risen in recent years. Although opioids provide modest relief, prolonged use of opioids is associated with several side effects, including drug addiction. Furthermore, many opioid trials do not sufficiently account for patient variability in the pathophysiology of pain. More recently, alternative methods to combat pain have emerged, including the use of opioid antagonists at low doses."

Reply: We have updated the following paragraph and included appropriate references. The modified text reads as follows:

"Frequently, multifaceted treatments are required to manage chronic pain [1]. Due to the prevalence of painful conditions, the clinical use of opioids has risen in recent years [2,3,4]  Although opioids provide modest relief, prolonged use of opioids is associated with several side effects, including drug addiction [5-6]. Furthermore, many opioid trials do not sufficiently account for patient variability in the pathophysiology of pain [7]. More recently, alternative methods to combat pain have emerged, including the use of opioid antagonists at low doses[8]."

  1. Cohen, S. P., Vase, L., & Hooten, W. M. (2021). Chronic pain: an update on burden, best practices, and new advances. Lancet (London, England), 397(10289), 2082–2097. https://doi.org/10.1016/S0140-6736(21)00393-7
  2. Gleber, R., Vilke, G. M., Castillo, E. M., Brennan, J., Oyama, L., & Coyne, C. J. (2020). Trends in emergency physician opioid prescribing practices during the united states opioid crisis. American Journal of Emergency Medicine, 38(4), 735–740. https://doi.org/10.1016/j.ajem.2019.06.011
  3. Rosa, J., & Burke, J. R. (2021). Changes in opioid therapy use by an interprofessional primary care team: a descriptive study of opioid prescription data. Journal of Manipulative and Physiological Therapeutics, 44(3), 186–195. https://doi.org/10.1016/j.jmpt.2021.01.003
  4. Essack, Y., & Stanfliet, J. (2016). Opioid abuse. Professional Nursing Today, 20(4), 20–21.
  5. Bennett, C. D. (2017). New jersey's opiod addiction health crisis. Md Advisor : A Journal for New Jersey Medical Community, 10(2), 5–6.
  6. Paul, S. M., & Allread, V. (2017). Opiod misuse, abuse and addiction part 2: opiod prescriber responsibilities and resources. Md Advisor : A Journal for New Jersey Medical Community, 10(1), 4–16.
  7. Gilron, I., Carr, D. B., Desjardins, P. J., & Kehlet, H. (2018). Current methods and challenges for acute pain clinical trials. Pain Reports, 4(3). https://doi.org/10.1097/PR9.0000000000000647
  8. Bruun-Plesner, K., Blichfeldt-Eckhardt, M. R., Vaegter, H. B., Lauridsen, J. T., Amris, K., & Toft, P. (2020). Low-dose naltrexone for the treatment of fibromyalgia: investigation of dose-response relationships. Pain Medicine (Malden, Mass.), 21(10), 2253–2261. https://doi.org/10.1093/pm/pnaa001

Point 4: “Insert two tables summarizing the main characteristics of the clinical studies describing the therapeutic potential of LDN in the treatment of arthritic and non-arthritic diseases.”

Reply: We have included the following tables as recommended by the reviewer.

Table 3: Low Dose Naltrexone therapeutic benefits in Arthritis

Disease

LDN benefits

Rheumatoid Arthritis & Seropositive arthritis

Reduced dispensing of medications (antirheumatic agents and NSAIDS)

Osteoarthritis

Reduction in pain and less dependence compared to opioids

Table 4: Low Dose Naltrexone therapeutic benefits in non-arthritic diseases

Disease

LDN benefits

Fibromyalgia

Reduces pain intensity and quality

Multiple Sclerosis

Reduces suppression of lymphocytes and improves quality of life

Crohn’s Disease

Wound healing, minimal adverse effects, improvement of symptoms

Depression and anxiety

Decreased levels of depression and anxiety,

Point 5: “Before Conclusions, please clarify the strengths and the limitations of this narrative review.”

Reply: We have included the following new section to the revised paper:

Strengths and Limitations: This is a mini literature review on the benefits of LDN in arthritis-related pain conditions and other pain conditions. LDN is an emerging treatment option, and this paper reviews some of the most recent evidence encouraging the use of LDN. At present, there is an insufficient number of double-blinded clinical studies that examine the utility of LDN in various chronic pain conditions to develop a definitive treatment plan for LDN. In addition, there is a very limited number of studies on the benefits of LDN in arthritis and firm conclusions are not yet possible. Our review merely provides a summary of relevant peer-reviewed published studies and we remain agnostic about the clinical efficacy of low-dose naltrexone. Yet, we are optimistic that additional fundamental and translational studies, as well as clinical trials may be informative and may establish both the mechanistic basis and efficacy of LND in the future. 

Point 6: “The manuscript should be based on presenting especially the latest evidence from the chosen topic. Therefore, update the references because too many articles in the list have been published for more than 10 years (1, 2, 11, 20, 26, 28, 35, 36, 44, 47), even 15-20 years  (3-5, 13-18, 21, 30, 31).”

Reply: We have updated our references. We included older references to provide appropriate historical context to current studies.

Point 7: “-What is reference 41??”

Reply: Wel have deleted this reference because it is not a peer-reviewed primary source.

Round 2

Reviewer 2 Report

Table 3 and Table 4 should be moved before the Strengths and Limitations section.  In addition, the two tables must include the characteristics of the studies based on which they were carried out, as shown in Table 2. It is one of my initial requests. Tables 1, 3, and 4 should be mentioned in the text, just as was done with Table 2.   Best wishes,

Author Response

Reviewer #2 Comments:

“This Table 3 and Table 4 should be moved before the Strengths and Limitations section.  In addition, the two tables must include the characteristics of the studies based on which they were carried out, as shown in Table 2. It is one of my initial requests. Tables 1, 3, and 4 should be mentioned in the text, just as was done with Table 2.”

Reply: We thank the reviewer for indicating that we did not yet fully address the referencing of Tables in the text. We have now cited all Tables (1 to 4) in the main text. As a result of the narrative flow and order of appearance, Table 2 was relabeled Table 3 and Table 3 is now relabeled Table 2.

Round 3

Reviewer 2 Report

The manuscript has been improved and I appreciate it can be published in its current form.